# Type One Protein Phosphatase 4aD Negatively Regulates Cotton (*Gossypium hirsutum*) Salt Tolerance by Inhibiting the Phosphorylation of Kinases That Respond to Abscisic Acid

**DOI:** 10.3390/ijms26083471

**Published:** 2025-04-08

**Authors:** Pengfei Cao, Miao Zhao, Jinxin Liu, Mingwei Du, Xiaoli Tian, Fangjun Li, Zhaohu Li

**Affiliations:** 1Engineering Research Center of Plant Growth Regulator, Ministry of Education & College of Agronomy and Biotechnology, China Agricultural University, Beijing 100193, China; caopengfei@cau.edu.cn (P.C.); miao_zhao_95@126.com (M.Z.); jxliu0220@163.com (J.L.); dumingwei@cau.edu.cn (M.D.); tianxl@cau.edu.cn (X.T.); lizhaohu@cau.edu.cn (Z.L.); 2State Key Laboratory of Plant Physiology and Biochemistry, China Agricultural University, Beijing 100193, China

**Keywords:** cotton, salt stress, phosphoproteome, phosphoproteins, dephosphorylation

## Abstract

Salinity is one of the major factors limiting the growth, development, and yield of cotton. Although the mechanisms of cotton tolerance to salt stress have been studied, the regulatory roles and mechanisms of protein kinases and phosphatases in cotton salt response remain poorly understood. Here, we identify Type One Protein Phosphatase 4aD (*GhTOPP4aD*), belonging to the Type One Protein Phosphatase (TOPP) family, as a negative regulator in cotton salt stress response. To reveal the post-translational modification mechanism by which *GhTOPP4aD* regulates salt stress response in cotton, phosphoproteome analysis was performed. A total of 6055 phosphoproteins with 12,608 phosphosites were identified. In VIGS-Ctrl plants, there were 935 upregulated and 35 downregulated phosphoproteins, while there were 1026 upregulated and 89 downregulated phosphoproteins in VIGS-*GhTOPP4aD* plants after NaCl treatment. Moreover, a class of tyrosine kinases responsive to abscisic acid (ABA) was significantly enriched at upregulated, differentially phosphorylated sites that were induced by NaCl in *GhTOPP4aD*-silenced plants, suggesting that these proteins could be regulated by dephosphorylation mediated by *GhTOPP4aD* in response to salt stress. Among them, Raf-like Kinase 36 (GhRAF36), FERONIA (GhFER), and Lysin Motif-containing Receptor-like Kinase 3 (GhLYK3) interacted with GhTOPP4aD and their kinase activities were inhibited by GhTOPP4aD. VIGS-*GhRAF36*, VIGS-*GhFER*, and VIGS-*GhLYK3* plants were sensitive to salt stress, suggesting that these kinases may play important roles in the regulation of cotton salt stress response mediated by GhTOPP4aD. These studies provide new insights into the mechanisms of cotton salt stress tolerance and the potential molecular targets for breeding salt-tolerant cotton varieties.

## 1. Introduction

Cotton stands out as a prominent fiber crop globally and plays a pioneering role in arid and saline-alkali soils [1]. Despite being categorized as salt-tolerant crop, it exhibits sensitivity to salt stress during the germination and seedling phases [2,3]. Irrigation water containing even trace levels of sodium chloride (NaCl) has the potential to elevate soil salinity [4]. When NaCl is taken up by the crop roots, the subsequent accumulation of salt within the crops leads to ionic toxicity and oxidative damage, impairing metabolic functions and decreasing photosynthetic efficiency [5,6]. Salt stress induces dehydration and wilting in plants, hinders cell growth and division, and disrupts various biochemical processes, ultimately leading to a decline in crop quality and yield [7,8]. Consequently, gaining insights into the molecular mechanisms underlying salt stress responses is pivotal for developing crops that can withstand salt stress.

The post-translational modification of proteins constitutes a significant field in protein chemistry research, playing a crucial role in plant growth and adaptation to both biotic and abiotic stresses [9,10,11]. Protein phosphorylation/dephosphorylation, as an important post-translational modification, is carried out by kinases and phosphatases [12]. Eukaryotic protein phosphatases are categorized into various types based on their catalytic mechanisms, substrate specificity, and sensitivity to inhibitors. These categories encompass serine/threonine-specific Phosphoprotein Phosphatase (PPP), Metal ion-dependent Protein Phosphatase (PPM), Dual Specificity Phosphatase (DSP), and Phosphotyrosine Phosphatase (PTP) [13]. Within the PPP family lies Protein Phosphatase 1 (PP1), which plays a pivotal role in diverse plant processes, including hormone regulation, growth and development, and light signaling [13,14,15]. For instance, Arabidopsis Type One Protein Phosphatase 4 (TOPP4) has been shown to dephosphorylate the DELLA protein in the gibberellin (GA) signaling pathway, facilitating the downstream transmission of GA signals and regulating plant growth and development in Arabidopsis [16]. TOPP4 also contributes to the development of leaf epidermal cells by modulating the phosphorylation state of PIN1 in Arabidopsis [17]. Additionally, Phytochrome B induces the phosphorylation of Phytochrome-interacting Factors 5 (PIF5), whereas TOPP4 dephosphorylates PIF5, enabling its accumulation and regulation of photomorphogenesis [18]. Among the PP1 family members, AtTOPP1 is identified to regulate the salt stress response by inhibiting the kinase activity of SnRK2.6 and the ABA signaling. [19]. However, the involvement of TOPPs in the salt stress response of cotton and the underlying mechanisms remain elusive.

Protein phosphorylation regulated by kinases primarily occurs on serine, threonine, and tyrosine residues, and it extensively participates in plant growth and development, stress responses, signal regulation, and more [20,21]. Plants respond to salt stress by triggering phosphorylation cascades. Salt stress induces an increase in cytosolic Ca^2+^ concentration. The EF-hand calcium-binding protein SOS3, upon sensing the calcium signal, interacts with SOS2 and phosphorylates SOS1, thereby maintaining ionic homeostasis within plant cells under salt stress [22,23]. Snf1-Related Protein Kinase2s (SnRK2s) are core components of the ABA signaling pathway, and the kinase activity of all members except SnRK2.9 can be activated by ABA and salt stress, participating in signal transduction in Arabidopsis under stress responses [24]. The receptor kinase FER negatively regulates the ABA signal but acts as a positive regulator of plant salt stress response. Upon activation, FER phosphorylates the phytochrome B (phyB) and decreases the abundance of phyB protein in the nucleus, thus promoting plant growth [25]. Arabidopsis RAF36 serves as a direct substrate of SnRK2s, thereby participating in ABA and stress responses [26]. Additionally, LYK3 is induced by ABA, and *atlyk3* mutants exhibit sensitivity to salt stress [27].

The rapid development of proteome and phosphoproteome profiling techniques in terms of throughput and sensitivity has provided new opportunities for the broader identification of phosphorylated proteins and sites. Quantitative proteomics based on isobaric tags for relative and absolute quantitation (iTRAQ) and tandem mass tags (TMTs) have also facilitated large-scale protein quantification [28,29,30]. Currently, phosphoproteome analysis has been widely used in the study of plant stress response mechanisms. In Arabidopsis, 468 regulated phosphopeptides exhibited significant changes under osmotic stress [31]. A total of 191 and 251 unique phosphopeptides, representing 173 and 227 phosphoproteins in two wheat cultivars (Hanxuan 10 and Ningchun 47), respectively, are identified that may play key roles in signal transduction and signaling cascades under drought stress [32]. Salt stress has been shown to induce significant changes in the phosphorylation levels of 189 phosphoproteins in sugar beet, and 15 differential phosphoproteins involved in signal transduction [33]. In okra, a total of 4341 phosphorylation sites have been identified in 2550 proteins, of which 91 sites were upregulated by NaCl treatment [34]. Although phosphoproteome analysis of plant responses to abiotic stress has been reported, phosphorylation modification in cotton under salt stress needs to be advanced. In this study, we identified GhTOPP4aD as a negative regulator of salt tolerance in cotton. By analyzing changes in protein phosphorylation levels involved in the early stages of salt stress in cotton using TMT label-based quantitative proteomics, we aimed to elucidate the molecular mechanisms of GhTOPP4aD regulating salt stress response in cotton.

## 2. Results

### 2.1. GhTOPP4aD Negatively Regulates Cotton Salt Response

The crucial regulatory mechanism for plants in response to stress signals involves post-transcriptional reversible phosphorylation modification, which is governed by protein kinases and phosphatases [35]. As one of the most significant classes of phosphatases in regulating stress responses, the specific roles of Type One Protein Phosphatases (TOPPs) in cotton salt stress response have yet to be uncovered. In Arabidopsis, AtTOPP4 plays a crucial role in diverse developmental processes, including light signal transduction, hormone regulation, and morphogenesis [16,18,36]. Nevertheless, its involvement in plant salt stress response, especially in cotton, has yet to be documented. By conducting a BLAST search in the *G. hirsutum* NBI proteins database (https://www.cottongen.org/blast, accessed on 22 February 2025), we pinpointed the cotton orthologs that exhibited the highest amino acid sequence similarity to Arabidopsis TOPP4. In terms of gene nomenclature, lowercase letters (a) appended to the cotton gene names serve to differentiate between genes within the same evolutionary branch. Meanwhile, the final uppercase letters A and D signify distinct subgenomes (Appendix A).

As a putative protein phosphatase in cotton, we performed an in vitro assay using pyro-nitrophenyl phosphate (pNPP) as the substrate to detect its protein phosphatase activity. Purified recombinant GST-GhTOPP4aD or GST proteins from *Escherichia coli* BL21 were incubated with pNPP. The GhTOPP4aD proteins demonstrated robust phosphatase activity by hydrolyzing pNPP into p-nitrophenol, a chromogenic product that absorbs light at 405 nm (Figure 1a).

To gain a deeper understanding of the biological function of GhTOPP4aD, we employed Virus-Induced Gene Silencing (VIGS) to silence *GhTOPP4aD* and observed that VIGS-*GhTOPP4aD* plants, in which gene expression was reduced by 66.33%, displayed a greater quantity of flatter and vigorous leaves compared to the VIGS-Ctrl plants (Figure 1b,c). In agreement with our observations, silencing *GhTOPP4aD* led to a significant increase in fresh weight and chlorophyll content by 41.09% and 26.27%, respectively, compared to the control plants under salt stress (Figure 1d,e). Preserving the equilibrium of the sodium-to-potassium ratio (Na^+^/K^+^) constitutes a crucial physiological trait for plants to endure salt stress [37]. Subsequently, we measured the Na^+^ and K^+^ contents in VIGS-*GhTOPP4aD* plants and found that silencing *GhTOPP4aD* markedly reduced the Na^+^ content by 56.21%, whereas there was no significant difference in K^+^ levels compared to those in VIGS-Ctrl plants (Appendix A), leading to a notably lower Na^+^/K^+^ ratio in VIGS-*GhTOPP4aD* leaves under salt stress conditions (Figure 1f). Salt stress induces hydrogen peroxide (H_2_O_2_) production in plants [38,39]. To assess the H_2_O_2_ levels in VIGS plants, we conducted a DAB staining assay. The results revealed that the leaves of VIGS-*GhTOPP4aD* plants exhibited lower H_2_O_2_ levels compared to the VIGS-Ctrl after salt stress (Figure 1g).

### 2.2. Overexpressing GhTOPP4aD Plants Are More Sensitive to Salt-Stress than the Recipient (HM1)

To clarify the function of *GhTOPP4aD* in cotton response to salt stress, we overexpressed *GhTOPP4aD* in the recipient plant (HM1) under the control of the 35S promoter and obtained two independent homozygous lines (1# and 7#). Western blotting analysis with anti-GhTOPP4aD antibodies showed substantially higher protein levels in OE-*GhTOPP4aD* than HM1 (Figure 2a). Contrary to VIGS-*GhTOPP4aD*, the OE-*GhTOPP4aD* plants, with gene expression levels increased by 26.84-fold and 22.23-fold, respectively, exhibited more sensitivity to salt stress with severe leaf wilting, less fresh weight of leaf, and chlorophyll content compared to VIGS-Ctrl plants (Figure 2b–e). Furthermore, the two lines overexpressing *GhTOPP4aD* exhibited a significant increase in Na^+^ content by 14.24% and 12.51%, respectively, as well as an elevation in the Na^+^/K^+^ ratio by 26.33% and 29.78% in leaves, compared to the recipient line HM1 (Figure 2f and Appendix A). DAB staining assays showed that the leaves of OE-*GhTOPP4aD* plants exhibited higher H_2_O_2_ levels compared to HM1 under salt stress (Figure 2g). These results demonstrated that GhTOPP4aD as a protein phosphatase negatively regulates cotton salt stress response.

### 2.3. Salt-Responsive Phosphosites and Phosphoproteins Modulated by GhTOPP4aD

To unravel the prospective molecular mechanism of *GhTOPP4aD* regulating cotton salt stress response as a protein phosphatase, we performed a phosphoproteome analysis with VIGS-*GhTOPP4aD* plants treated with NaCl (Appendix A). A total of 14,991 phosphopeptides were identified, of which 13,599 were modified peptides with 17,919 phosphosites corresponding to 6810 phosphoproteins (Table 1). In order to ensure the high reliability of the results, we used the localization probability > 0.75 standard to filter the authentication data. A total of 12,608 phosphosites corresponding to 6055 phosphoproteins were identified (Table 1), among which phosphorylation was situated primarily on serine (88.43%), and to a lesser extent, on threonine (11.16%) or tyrosine residues (0.41%) (Figure 3a). Moreover, most phosphopeptides (11,733 or 96.55%) had a single phosphosite, and only 3.21% had two phosphorylation sites (Figure 3b). More than half (3409 or 56.30%) of the phophoproteins had a single phosphosite, and 1240 (20.48%), 586 (9.68%), and 312 (5.15%) phosphoproteins had two, three, and four phosphosites, respectively (Figure 3c).

Principal components analysis (PCA) was carried out to evaluate the variability of multiplexed phosphoproteome samples. Four groups (VIGS_Ctrl under normal conditions, VIGS_*GhTOPP4aD* under normal conditions, VIGS_Ctrl under salt treatment, and VIGS_GhTOPP4aD under salt treatment) were separated well in the PCA plots, suggesting reproducible differences present among the different groups (Figure 3d).

Considering that there were two biological replicates in this study, the standard coefficient of variation (*CV*) of each phosphosite in the comparison group was set as a significance indicator. When the fold change of each phosphosite in the comparison group was > 1.3 or < 0.77 and *CV* < 0.1, the phosphosite was considered to have a significant change in the comparison group. According to this standard, the data showed that 1112 upregulated phosphosites corresponded to 935 phosphoproteins and 36 downregulated phosphosites corresponded to 35 phosphoproteins in control plants after salt treatment. In VIGS-*GhTOPP4aD* plants, 1264 upregulated phosphosites corresponded to 1026 phosphoproteins and 100 downregulated phosphosites corresponded to 89 phosphoproteins after NaCl treatment (Figure 3e). In general, the number of upregulated phosphosites and phosphoproteins were much higher than that of downregulated ones, consistent with the expectation that silencing protein phosphatase gene could show upregulation of protein phosphorylation.

### 2.4. GhTOPP4aD Regulates Protein Phosphorylation Levels in Diverse Biological Processes and Pathways

We further de-duplicated the identified differential phosphorylation sites and obtained 2206 differential phosphorylation sites (Appendix A). In order to determine the (de-) phosphorylation pattern of these phosphosites in different samples, the Mfuzz method was used to perform expression pattern cluster analysis, and the phosphosites with significant changes were divided into six clusters (Figure 4). Among all the clusters, only cluster 4 had downregulated phosphorylation after NaCl treatment, and phosphoproteins in cluster 4 were enriched in plant hormone signal transduction pathway and in the biological process of potassium ion transmembrane transport (Appendix A). Compared with normal conditions, the phosphorylation levels of cluster 1 and 2 in both VIGS-Ctrl and VIGS- *GhTOPP4aD* plants were upregulated after salt treatment, but the upregulation level in VIGS-*GhTOPP4aD* plants was lower than that in VIGS-Ctrl plants. Unlike cluster 2 enrichment in transport-related biological processes, cluster 1 enrichment was in the protein polyubiquitination and auxin-activated signaling pathways (Appendix A). The phosphorylation levels of cluster 3 were similar in control and VIGS-*GhTOPP4aD* plants both under normal conditions and after NaCl treatment, indicating that the proteins in cluster 3 only respond to salt stress (Appendix A). After NaCl treatment, the phosphorylation levels of cluster 5 and 6 were significantly upregulated in the VIGS-*GhTOPP4aD* plants compared with the control plants and the phosphorylation levels of the proteins in cluster 6 were also higher in VIGS-*GhTOPP4aD* plants without NaCl treatment. The differentially phosphorylated proteins in cluster 6 were mainly enriched into the starch and sucrose metabolism and ABA signaling pathways (Figure 4, Appendix A). Intriguingly, we found that Abscisic Acid Insensitive5 (ABI5), a positive regulator of the ABA signaling pathway, was also among these differentially phosphorylated proteins (Figure 4, Table 2).

### 2.5. GhTOPP4aD Regulates the Dephosphorylation of Numerous Kinases Under Salt Stress

The Venn diagram showed that NaCl treatment significantly induced the phosphorylation of a large number of sites both in VIGS-Ctrl and VIGS-*GhTOPP4aD* plants. Removing 289 phosphosites modulated by salt treatment, 975 upregulated phosphosites that specifically regulated by GhTOPP4aD under salt stress were identified as the putative targets for GhTOPP4aD dephosphorylation (Figure 5a).

Domain enrichment analysis was performed on 975 upregulated phosphosites, and 70 enriched conserved domains were obtained, including protein kinases, transcription factors, and other domains (Figure 5b). The protein tyrosine and serine/threonine kinase domains were enriched with 36 proteins (Appendix A), including GhRAF36, GhFER, GhLYK, and GhSRF, which related to ABA signaling pathway. Under salt stress, silencing *GhTOPP4aD* resulted in significantly upregulated phosphorylation levels of these kinase proteins, compared to that in Ctrl plants. (Figure 5c–f). Moreover, the upregulated phosphorylation sites were identified as Serine (S) 131 residue on GhRAF36 (Figure 5c), S693 on GhSRF6 (Figure 5d), S902 on GhFER (Figure 5e), and S304 on GhLYK3 (Figure 5f). These results indicated that *GhTOPP4aD* regulates salt stress response may through the dephosphorylation of different kinases.

### 2.6. GhTOPP4aD and the Candidate Tyrosine Kinases Interact In Vivo

Next, we conducted Yeast Two-Hybrid (Y2H) assays to detect the interaction between GhTOPP4aD and the candidate kinases. The results showed that GhRAF36 and GhFER strongly bind to GhTOPP4aD (Figure 6a,b), while GhLYK3 and GhSRF6 exhibited weaker interactions with GhTOPP4aD (Figure 6c,d). Additionally, we also examined tyrosine kinases GhHERK1 and GhLRK, which exhibited no significant difference in phosphorylation levels, and used these two kinases as negative controls. The interaction between GhHERK1 or GhLRK and GhTOPP4aD is almost undetectable (Appendix A). To further confirm the interactions between GhTOPP4aD and the kinases, we performed a Luciferase Complementation Imaging (LCI) assay, using GhOST1-nLUC and cLUC-GhABI1 as positive controls. Consistently, GhRAF36, GhFER, GhLYK3, and GhSRF6 also interacted with GhTOPP4aD in tobacco (Figure 6e–h). However, no interaction was observed between GhTOPP4aD and GhHERK1 or GhLRK in tobacco (Appendix A).

### 2.7. In Vitro Dephosphorylation Assay to Analyze the Effect of GhTOPP4aD on Tyrosine Kinase Activity

Considering the interaction with GhTOPP4aD may regulate the phosphorylation of these tyrosine kinases, we conducted in vitro dephosphorylation assays with recombinant MBP-GhRAF36, MBP-GhFER, MBP-GhLYK3, and MBP-GhSRF6 as substrates. Western blot results revealed that GhTOPP4aD strongly inhibited the kinase activity of GhRAF36 and GhFER in a dose-dependent manner (Figure 7a,b), and exhibited weaker inhibitory effects on GhLYK3 kinase activities (Figure 7c). In contrast, the kinase activities of GhSRF6, GhHERK1, and GhLRK were almost unaffected by GhTOPP4aD (Figure 7c and Appendix A). Taken together, these results suggested that GhTOPP4aD may directly interact with GhRAF36, GhFER, and GhLYK3 to inhibit their kinase activities, thereby negatively regulating salt stress response in cotton.

### 2.8. Silencing GhTOPP4aD Enhances NaCl-Induced Phosphorylation of GhRAF36, GhLYK3, and GhFER Kinases

Furthermore, we performed in vitro phosphorylation assays to check whether NaCl could induce the phosphorylation of the candidate kinases. Due to the strong autophosphorylation activity of tyrosine kinases, we pretreated these kinases with Calf Intestine Phosphatase Alkaline (CIPA) to serve as substrates and the total proteins extracted from VIGS-Ctrl and VIGS-*GhTOPP4aD* plants with NaCl treatment as the kinases. The results showed that salt could induce the phosphorylation of GhRAF36 and GhLYK3, and silencing GhTOPP4aD further enhanced the phosphorylation (Figure 8a,b). Recent studies have demonstrated that salt stress inhibits the kinase activity of FER [25]. Consistently, we also noted that salt hindered the phosphorylation of GhFER in cotton, while silencing *GhTOPP4aD* counteracted the inhibitory effect of salt stress on GhFER (Figure 8c). Together, these results suggested that GhTOPP4aD interacts with and inhibits the autophosphorylation of GhRAF36, GhFER, and GhLYK3 by directly dephosphorylation.

### 2.9. Silencing GhRAF36, GhFER, and GhLYK3 Enhance Cotton Salt Stress Sensitivity

To elucidate the functions of candidate kinases in response to salt stress, we silenced them in cotton using VIGS. Compared to the VIGS-Ctrl, the plants with gene expression levels reduced by 60.67%, 73.01%, and 55% for *GhRAF36*, *GhFER,* and *GhLYK3 genes*, respectively, exhibited severe wilting and more wilt spots in the leaves, indicating their sensitivity to salt stress (Figure 9a–f). Additionally, VIGS-*GhSRF6* plants with gene expression levels reduced by 45.33% (Appendix A), VIGS-*GhHERK1* plants with gene expression levels reduced by 68.33% (Appendix A), and VIGS-*GhLRK* plants with gene expression levels reduced by 35.67% (Appendix A), showed no significant phenotypic differences compared to the VIGS-Ctrl plants with or without salt stress. These results suggest that *GhRAF36*, *GhLYK3*, and *GhFER* are positive regulators of the salt stress response in cotton, and that the functions of *GhLYK3* and *GhFER* are conserved between Arabidopsis and cotton [27,40,41].

## 3. Discussion

Reversible protein phosphorylation serves as a crucial regulatory mechanism for plants to cope with biotic and abiotic stresses [35,42]. Although studies on the mechanisms of cotton salt tolerance have been conducted, the complex phosphorylation regulatory mechanisms under salt stress remain to be fully explored. In this study, our results indicate that *GhTOPP4aD* functions as a negative regulator of cotton salt stress response. VIGS-*GhTOPP4aD* plants exhibited significantly enhanced resistance to salt stress compared to VIGS-Ctrl plants (Figure 1), while overexpression of *GhTOPP4aD* rendered plants sensitive to salt stress (Figure 2). To delve into the mechanism by which *GhTOPP4aD* regulates cotton salt stress response, we conducted comprehensive proteomic and phosphoproteome analyses on VIGS plant seedlings with or without NaCl treatment, quantifying 6055 proteins, 13,599 distinct phosphopeptides, and 12,608 phosphorylation sites (Table 1). Compared to VIGS-Ctrl, 975 phosphoproteins were differentially upregulated in VIGS-*GhTOPP4aD* plants, including protein tyrosine and serine/threonine kinase involved in ABA and salt stress response (Figure 4 and Figure 5), which are of great significance for us to elucidate the mechanisms of cotton salt stress response.

Elevated salinity levels in plants lead to osmotic stress and Na^+^ ions toxicity [43], prompting an increase in the plant hormone ABA [44,45]. ABA swiftly activates the expression of genes that respond to stress and initiates various physiological reactions, such as stomatal closure to conserve water in plants [46,47,48]. Genetic and molecular studies have uncovered several components involved in ABA response. For instance, bZIP-type transcription factor ABI5 is discovered through analysis of the *abi5-1* mutant that exhibits recessive insensitivity to ABA [49]. ABI5, predominantly expressed in seeds and highly induced by ABA, is crucial for regulating seed germination and the initial growth phase of seedlings [42,49,50,51,52]. Here, we identified a class of abscisic acid-activated signaling pathway proteins that exhibited an up-phosphorylation pattern in VIGS-*GhTOPP4aD* plants with NaCl treatment, including GhABI5 (Figure 4, Table 2 and Appendix A). Previous studies have shown that ABI5 is tightly regulated post-translationally. For instance, the protein kinases SnRK2.2, SnRK2.3, and SnRK2.6 have been demonstrated to phosphorylate and stabilize the ABI5 protein in response to ABA [53,54,55,56,57]. Conversely, the phosphatase PROTEIN PHOSPHATASE6 reverses the phosphorylation of the ABI5 protein [58]. In the presence of ABA, the Brassinosteroids (BRs) signaling repressor Brassinosteroid Insensitive 2 (BIN2) can stabilize and phosphorylate ABI5, which acts as a positive regulator of ABA, thereby positively modulating the ABA response. The antagonism between BR and ABA may be mediated by the inactivation of the BIN2-ABI5 cascade when BRs are applied to plants [59]. In this study, we observed an upregulation of GhABI5 phosphorylation levels in VIGS-*GhTOPP4aD* plants treated with NaCl, which may be one of the reasons for GhTOPP4aD-mediated negative regulation of cotton salt tolerance. Previous studies have shown that the ABI5 protein has at least eight phosphorylation sites [56,57,60]. However, only the Ser309 of GhABI5 were detected in our phosphoproteome data (Table 2 and Appendix A). Therefore, Ser309 may be a site regulated by GhTOPP4aD. However, the kinase responsible for phosphorylating GhABI5 in VIGS-*GhTOPP4aD* plants remains unclear, which will be an area of research in our future work.

We also observed a significant enrichment of protein tyrosine kinases in specific protein domains, and the phosphorylation levels of these kinases changed in response to salt stress (Figure 5, Table 2 and Appendix A). Interestingly, FER, RAF36, and LYK3 kinase are known to regulate ABA signaling pathways or abiotic stress responses in Arabidopsis were also included [26,27,40,61]. This suggests that these tyrosine kinases may be targets of GhTOPP4aD. To test this hypothesis, we analyzed the phenotypes of these tyrosine kinases and found that the VIGS-*GhFER*, VIGS-*GhRAF36*, and VIGS-*GhLYK*3 plants were more sensitive to salt stress compared to VIGS-Ctrl plants (Figure 9). Moreover, GhFER, GhRAF36, and GhLYK3 interact with GhTOPP4aD, and their kinase activities were inhibited by GhTOPP4aD (Figure 6 and Figure 7). We also found that silencing GhTOPP4aD further enhanced the phosphorylation of GhFER, GhRAF36, and GhLYK3 that were induced by NaCl (Figure 8). These results indicated that GhTOPP4aD may regulate cotton salt tolerance by affecting the activities of these tyrosine kinases.

Prior research has shown that the B2, B3, and B4 subfamilies of Raf-like protein kinases serve as upstream kinases that phosphorylate and activate SnRK2s, playing a crucial role in mediating osmotic stress and ABA responses [62]. A recent study revealed that two members of the B1 subfamily, RAF13 and RAF15, can also act as upstream kinases to phosphorylate and activate SnRK2s, as evidenced by the impaired ABA-triggered SnRK2.6 activation and stomatal closure observed in *raf15* mutants [63]. In addition, *GhRAF42,* a member of the B subfamily in cotton, has been identified as a positive regulatory factor in response to salt stress, and VIGS-*GhRAF42* plants are sensitive to salt stress [64]. We speculate that silencing *GhTOPP4aD* may relieve its inhibitory effect on the kinase activity of GhRAF36, allowing activated GhRAF36 to potentially phosphorylate and activate SnRK2s or downstream transcription factors in the ABA signaling pathway, thereby enhancing cotton’s resistance to salt stress. This provides a new insight into the mechanisms of cotton salt tolerance and is a question that warrants further investigation in the future.

As a positive regulator of salt stress, FER plays a significant role in plant salt tolerance [25,40,41]. Studies have shown that FER regulates plant salt stress by maintaining the integrity of the cell wall [40]. Recently, phosphorylation of phyB mediated by FER has been found to facilitate the dissociation of photobodies and reduce the abundance of phyB protein in the nucleus, thereby promoting plant growth. However, salt stress inhibits FER kinase activity, leading to enhanced photobody stability and increased abundance of non-phosphorylated phyB protein in the nucleus. This, in turn, promotes the degradation of PIFs, resulting in impaired plant growth [25]. In this study, the kinase activity of GhFER was also inhibited by salt stress but enhanced after *GhTOPP4aD* silencing (Figure 8 and Figure 9), and VIGS-*GhTOPP4aD* plants exhibited more flattened and vigorous leaves compared to VIGS-Ctrl plants (Figure 1b), suggesting that GhTOPP4aD may negatively regulate cotton salt tolerance via inhibiting GhFER kinase activity. In summary, these results will provide some evidence for clarifying the negative regulator role of GhTOPP4aDin cotton salt stress response and suggest potential molecular targets for selecting and breeding salt-tolerant cotton varieties.

## 4. Materials and Methods

### 4.1. Plant Materials and Growth Conditions

*Gossypium hirsutum* (L.) XinShi 17 was maintained in our laboratory. The cotton transgenic recipient plants HM1 and OE-*GhTOPP4aD* plants were generously supplied by Wuhan Towin Biotechnology Company Limited, Wuhan, China. Cotton seeds were germinated in sand and subsequently transplanted into Hoagland nutrient solution five days (d) after sowing (DAS). OE-*GhTOPP4aD* and XinShi 17 seedlings prepared for injection were cultivated under controlled conditions of 24 °C, 60% relative humidity, and a light intensity of 400 μmol m^−2^s^−1^ with a 14-h light/10-h dark photoperiod. Additionally, *Nicotiana benthamiana* plants were grown in pots containing a 1:1 mixture of soil and vermiculite (*w*/*w*) in a controlled growth chamber maintained at 22 °C, 60% relative humidity, and a light intensity of 80 μmol m^−2^s^−1^ under the same 14-h light/10-h dark photoperiod.

### 4.2. Virus-Induced Gene Silencing (VIGS) Assay

The VIGS assay was carried out following a previously established protocol [65]. In summary, plasmids including *pTRV-RNA1* or *pTRV-RNA2* (Ctrl, *GhCLA1*, and *GhTOPP4aD* or other genes) were introduced into the *Agrobacterium tumefaciens* strain GV3101. The *Agrobacterial* culture was incubated overnight at 28 °C in YEP liquid medium supplemented with 50 μg/mL kanamycin, 25 μg/mL gentamicin, 10 mM MES, and 20 μM acetosyringone. Subsequently, the cells were centrifuged at 8000 rpm for 5 min at room temperature to form a pellet, which was then resuspended in infiltration buffer containing 10 mM MgCl_2_, 10 mM MES, and 200 μM acetosyringone. A mixture of *pTRV-RNA2* at an OD_600_ of 1.5 and pTRV-RNA1 in a 1:1 ratio was infiltrated into two fully expanded cotyledons of three-day-old plants grown hydroponically using a needleless syringe. The VIGS-*GhCLA1* plants served as indicators to assess the reliability of genes.

### 4.3. Real-Time Quantitative PCR (RT-qPCR)

The total RNA was extracted from VIGS plants using the Plant RNA Mini Kit (Aidlab, Beijing, China), and 2 μg RNA was utilized for synthesizing first-strand cDNA with the first-strand cDNA synthesis kit (Aidlab, Beijing, China). The synthesized cDNA was then subjected to qRT-PCR using iTaq SYBR green Supermix (Bio-Rad, Shanghai, China) on an Applied Biosystems 7500 Fast real-time PCR system, adhering to standard protocols. *GhActin9* served as the internal reference gene. The primers employed for qRT-PCR can be found in Appendix A.

### 4.4. Measurement of Sodium and Potassium Ion Content

To determine the sodium and potassium ion content, start by grinding the dried plant samples into a fine powder. Precisely weigh out 0.2 g powder and transfer it into a 10 mL centrifuge tube. Next, add 5 mL of 1 mol/L HCL to extract the ions. Place the tube in a shaker and allow it to oscillate overnight at 28 °C. After that, filter the extract through a 9 cm qualitative filter paper. Subsequently, dilute the filtered solution with HCL to a total volume of 5 mL. Finally, measure the absorbance of the diluted solution using a flame spectrophotometer to assess the levels of sodium and potassium ions.

### 4.5. Determination of Chlorophyll Content

Leaf samples weighing approximately 0.2 g were collected from fourteen-day-old plants and immersed in 80% acetone to effectively extract the pigments. Subsequently, spectrophotometric analysis was performed on the extracted solution at wavelengths of 663 nm and 645 nm.

### 4.6. Diaminobenzidine (DAB) Staining

To visualize the accumulation of H_2_O_2_, leaves were first washed with deionized water and then submerged in a 50 mM Tris-HCl buffer (pH 3.8) containing 1 mg/mL of DAB. The leaves were then subjected to vacuum infiltration for 30 min to facilitate penetration of the staining solution. Following this, they were incubated at 25 °C in the dark for 24 h. To finalize the staining process, the leaves were immersed in a boiling water bath containing a 9:1 mixture of ethanol and glycerol until they were completely cleared of chlorophyll.

### 4.7. Measurements of Phosphatase Activity

The phosphatase activity of GhTOPP4aD was evaluated by reacting the recombinant GST-GhTOPP4aD protein with 1 mM Pyronitrophenyl phosphate (pNPP), serving as the substrate, in a phosphatase buffer composed of 50 mM Tris-HCl at pH 7.0 and 2 mM DTT. The absorbance of the reaction mixture was then determined at a wavelength of 405 nm.

### 4.8. Recombinant Protein Expression and Purification

The CDS of the relevant genes were individually cloned into the pET28a vector (His tag), pGEX4T-1 vector (GST tag), and the pMAL-c2X vector (MBP tag). These constructed vectors were then transformed into *E. coli* BL21 cells. To induce the expression of the recombinant proteins fused with these different tags, 0.2 mM IPTG was added to the cultures, which were then incubated at 18 °C for 14–16 h. Subsequently, the cultures were harvested by centrifuging at 6000× *g* for 3 min at 4 °C. The recombinant proteins were purified using Ni Sepharose resin (17371202, GE Healthcare, Shanghai, China) for the His tag, with Amylose Resin High Flow (BioLabs, Shanghai, China, #E8022) for the MBP tag, or Pierce™ Glutathione Agarose resin (Thermo Fisher Scientific, Waltham, MA, USA) for the GST tag, respectively.

### 4.9. Detecting GhTOPP4aD Protein Abundance

Fourteen-day-old OE-*GhTOPP4aD* were collected and immediately frozen in liquid nitrogen. Proteins were extracted with extraction buffer (25 mM Tris–HCl, pH 7.5, 10 mM NaCl, 10 mM MgCl_2_, 0.5% Tween 20, 1 mM EDTA, 1 mM DTT, and 4 mM PMSF), and an abundance of GhTOPP4aD was determined using anti-GhTOPP4aD antibodies (provided by Nanjing GenScript Biotech Corporation, Nanjing, China) by Western blotting analysis.

### 4.10. Yeast Two-Hybrid Assay

The full-length sequences of *GhTOPP4aD* and the gene to be validated were amplified and cloned into pGADT7 and pGBKT7 vectors, respectively. Various pairs of these plasmids were co-transformed into Y2H-Gold yeast cells. The transformed yeast cells were then plated onto agar media with different synthetic dropout formulations: SD/-Trp-Leu, SD/-Trp-Leu-His supplemented with 10 mmol/L 3-amino-1,2,4-triazole (3-AT), or SD/-Trp-Leu-His-Ade for 4–5 days to assess protein interactions. The primer sequences utilized for the cloning process are provided in Appendix A.

### 4.11. Luciferase Complementation Imaging (LCI) Assay

The LCI assays were conducted as previously described [66]. In brief, the coding sequences (CDS) of *GhTOPP4aD* and the gene to be validated were cloned into either pCAMBIA1300-nLUC or pCAMBIA1300-cLUC vectors. These constructs were then co-transformed into *Agrobacterium tumefaciens* strain GV3101 and co-infiltrated into the leaves of *Nicotiana benthamiana* plants. After an incubation period of 48–72 h, the luciferase (LUC) signal was detected using a cold charge-coupled device camera. The primers employed for the LCI assay are provided in Appendix A.

### 4.12. In Vitro Dephosphorylation Assay

The purified recombinant proteins of MBP-kinases (2–3 µg) were incubated with His-GhTOPP4aD (1–3 µg) in phosphatase buffer (25 mM Tris-HCl at pH 7.4, 12 mM MgCl_2_, and 2 mM DTT) at 30 °C for 30 min. Following the incubation, 5 × SDS sample buffer was added to the reaction mixture, which was then boiled for 10 min. Subsequently, the samples were resolved using 12.5% SDS-PAGE. The phosphorylation status was assayed using a pIMAGO-biotin phosphoprotein detection kit, as described in previous studies [67,68].

### 4.13. In Vitro Kinase Assay

About 0.2 g of leaves from fourteen-day-old VIGS plants were homogenized in 200 μL of total protein extraction buffer (50 mM Tris-HCl at PH7.4, 150 mM NaCl, 0.5 mM EDTA, 5% Glycerol, 2.5 mM MgCl_2_, and 1% Triton X-100). The total protein extracted from these VIGS plants was utilized to phosphorylate pre-dephosphorylated recombinant proteins (bound to beads) with Calf Intestine Phosphatase Alkaline (CIPA) in an 80 μL reaction buffer (25 mM Tris-HCl, pH 7.4, 5 mM MgCl_2_, 50 μM ATP, and 1 mM DTT) at 30 °C for 30 min. Afterward, the phosphorylated proteins (bound to beads) were collected by centrifuging and washed three times with kinase buffer. The reactions were terminated by adding 5×SDS loading buffer following the incubation period. Finally, the phosphorylation was detected using a pIMAGO-biotin phosphoprotein detection kit.

### 4.14. Phosphoproteome Analysis

About 2 g samples of 2nd true leaves from two-week-old seedlings of VIGS-Ctrl and VIGS-*GhTOPP4aD* treated with normal growth and 300 mM NaCl for 30 min were collected and frozen in liquid nitrogen, then sent to PTM-BioLab Company (Hangzhou, China) for protein extraction, trypsin digestion, labeling, high performance liquid chromatography (HPLC) fractionation, affinity enrichment, and liquid chromatography-tandem mass spectrometry (LC–MS/MS) analysis. Secondary mass spectrometry data were retrieved using Maxquant (v1.6.15.0) to blast in the database Blast_Gossypium_hirsutum_3635_PR_20210112. The enzyme digestion method was set as Trypsin/P, the number of missing cuts was set to two. The minimum length of the peptide was set to seven amino acid residues, and the maximum number of peptide modifications was set to five. The mass error tolerance for primary parent ions was set at 20 ppm and 4.5 ppm, respectively, for First search and Main search, and 20 ppm for the precursor mass tolerance. Oxidation (M), n-terminal acetylation of protein, deamidation (NQ), and phosphorylation of Serine (S), Threonine (T), and Tyrosine (Y) was defined as variable modifications, while Carbamidomethyl (C) was set as a fixed modification. The quantitative method was set to MTT-11plex, and the FDR for protein identification and PSM identification was set to 1%.

Principal component analysis (PCA) was carried out to determine the consistency of two biological replicates using OmicStudio tools (https://www.omicstudio.cn/tool, accessed on 22 February 2025) [69]. For each peptide, its phosphorylation level in different samples were normalized and then used for the analysis of PCA. To determine the differential phosphorylation modification sites between the two samples under different treatments, the cutoff values of *CV* < 0.1 and fold change (FC) > 1.3 or FC < 1/1.3 were adopted. The expression pattern cluster analysis was carried out by PTM-BIO Shiny Tool. The relative phosphorylation levels of phosphosites were transformed by Log2 and analyzed by the Mfuzz method. The biological process of Gene Ontology (GO), Kyoto Encyclopedia of Genes and Genomes (KEGG), and domain enrichment analysis of phosphoproteins were analyzed by Fisher’s accurate test method, and terms with *p* < 0.05 were considered to be significantly enriched. The partially significant terms in GO and KEGG enrichment analyses were selected, while the top 20 significant terms in domain enrichment analysis were shown.

## Figures and Tables

**Figure 1 ijms-26-03471-f001:**
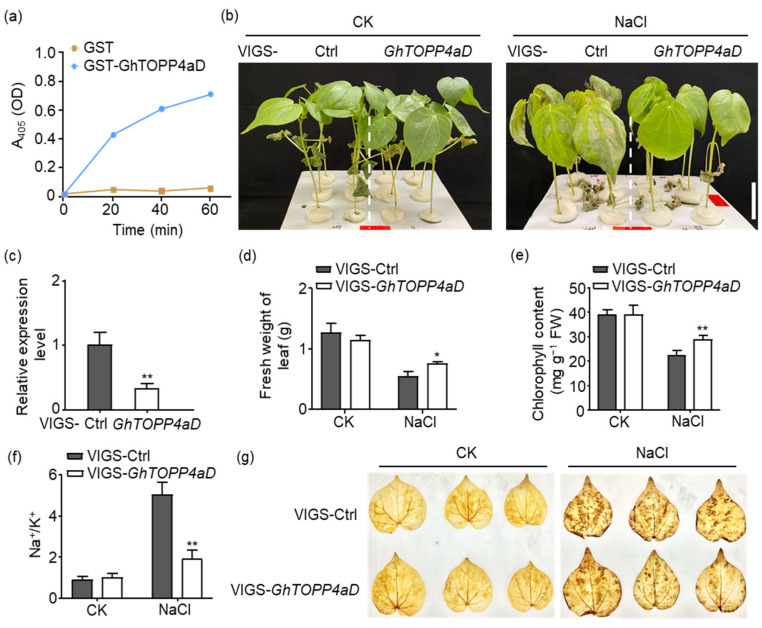
Silencing of *GhTOPP4aD* by virus-induced gene silencing (VIGS) confers salt-stress resistance in cotton. (**a**) Phosphatase activity of GhTOPP4aD is enhanced with increasing time points. (**b**) Silencing of *GhTOPP4aD* by VIGS is more salt-stress tolerant than the VIGS-Ctrl. “CK” represents control plants without NaCl stress; “NaCl” represents plants with 300 mM NaCl treatment. VIGS-Ctrl or genes refer to cotton plants that have been injected with *Agrobacterium tumefaciens* carrying the VIGS empty vector or gene-specific vector, respectively. The dashed lines separate different VIGS groups. Bar = 3 cm. (**c**) The relative expression level of *GhTOPP4aD* in VIGS plants. *GhActin9* was used as the internal control. The data are shown as means ± SD from three independent repeats (n = 3; **, *p* < 0.01, Student’s *t*-test). (**d**) The fresh weight of leaves of VIGS-Ctrl and VIGS-*GhTOPP4aD* with or without NaCl treatment. The data are shown as means ± SD from three independent repeats (n = 3; *, *p* < 0.05, Student’s *t*-test). (**e**) The chlorophyll content of VIGS-Ctrl and VIGS-*GhTOPP4aD* with or without NaCl treatment. The data are shown as means ± SD from three independent repeats (n = 3; **, *p* < 0.01, Student’s *t*-test). (**f**) Na^+^/K^+^ of VIGS-Ctrl and VIGS-*GhTOPP4aD* with or without NaCl treatment. The data are shown as means ± SD from three independent repeats (n = 3; **, *p* < 0.01, Student’s *t*-test). (**g**) Histochemical localization of H_2_O_2_ in VIGS-*GhTOPP4aD* cotton leaves with or without salt stress.

**Figure 2 ijms-26-03471-f002:**
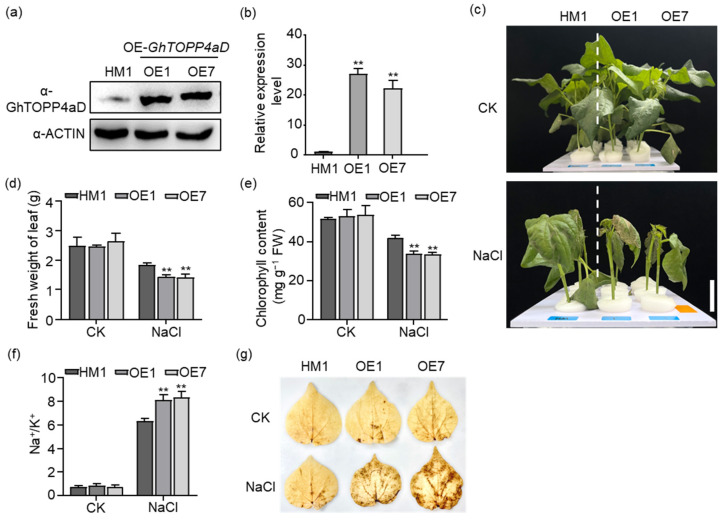
Overexpression of *GhTOPP4aD* in Cotton reduces salt tolerance. (**a**) GhTOPP4aD protein levels in HM1, transgenic lines 1# and 2# determined by immunoblotting using Anti-GhTOPP4aD antibodies. GhACTIN was used as a loading control. (**b**) The relative expression level of *GhTOPP4aD* in HM1 and transgenic plants. *GhActin9* was used as the internal control. The data are shown as means ± SD from three independent repeats (n = 3; **, *p* < 0.01, Student’s *t*-test). (**c**) OE-*GhTOPP4aD* plants are more salt-stress sensitive than the HM1. “CK” represents control plants without NaCl stress while “NaCl” represents plants with 300 mM NaCl treatment. Bar = 3 cm. (**d**) The fresh weight of leaves of HM1 and OE-*GhTOPP4aD* with or without NaCl treatment. The data are shown as means ± SD from three independent repeats (n = 3; **, *p* < 0.01, Student’s *t*-test). (**e**) The chlorophyll content of HM1 and OE-*GhTOPP4aD* with or without NaCl treatment. The data are shown as means ± SD from three independent repeats (n = 3; **, *p* < 0.01, Student’s *t*-test). (**f**) Na^+^/K^+^ of HM1 and OE-*GhTOPP4aD* with or without NaCl treatment. The data are shown as means ± SD from three independent repeats (n = 3; **, *p* < 0.01, Student’s *t*-test). (**g**) Histochemical localization of H_2_O_2_ in HM1 and OE-*GhTOPP4aD* cotton leaves with or without salt stress.

**Figure 3 ijms-26-03471-f003:**
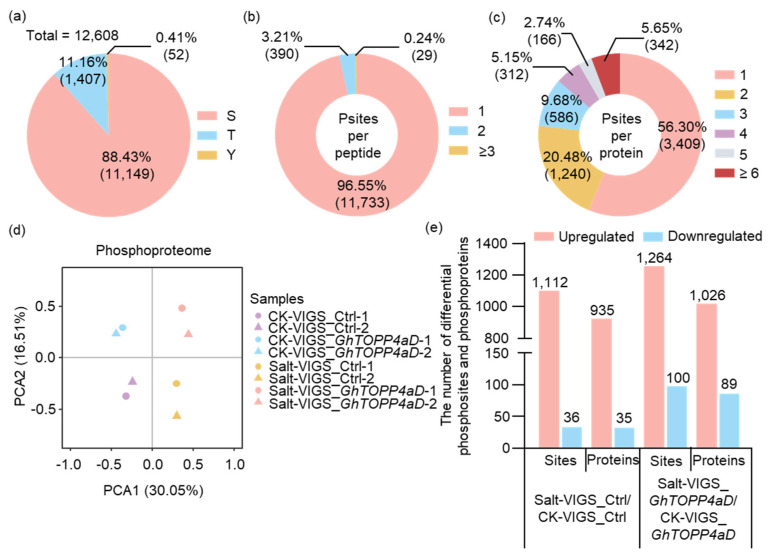
Overview of the phosphoproteome in VIGS-Ctrl and VIGS-*GhTOPP4aD* plants in response to 300 mM NaCl. (**a**) The percentage of phosphosites containing Ser (S), Thr (T), and Tyr (Y), and their numbers are shown in parentheses. (**b**) The percentage of phosphopeptides containing one, two, and no less than three phosphosites, and their numbers are shown in parentheses. (**c**) The percentage of phosphoproteins containing one, two, three, four, five, and more than six phosphosites, and their numbers are shown in parentheses. (**d**) Principal component analysis (PCA) of phosphoproteome data obtained from VIGS-Ctrl and VIGS-*GhTOPP4aD* plants under normal (CK) and 300 mM NaCl conditions. (**e**) The number of differential phosphosites and phosphoproteins in VIGS-Ctrl and VIGS-*GhTOPP4aD* plants under normal and 300 mM NaCl conditions.

**Figure 4 ijms-26-03471-f004:**
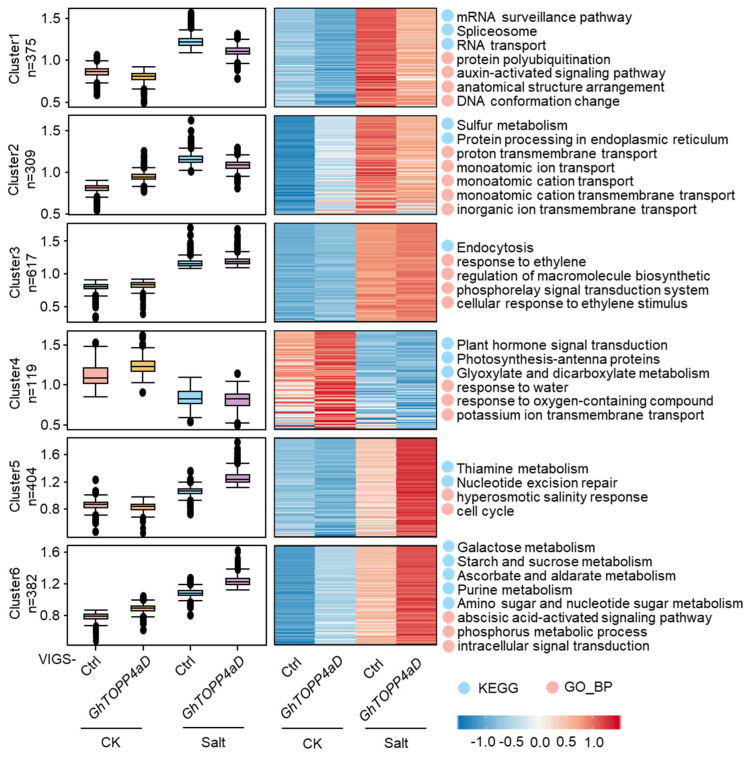
The expression and functional analysis of the differential phosphoproteins in VIGS-Ctrl and VIGS-*GhTOPP4aD* plants in response to 300 mM NaCl. The expression patterns of 2,206 differential phosphosites were analyzed by Mfuzz method. On the left is a box plot showing phosphorylation levels of phosphosites in different samples. In the middle is the expression heatmap. The relative phosphorylation level of phosphorylation sites is transformed by Log2. The phosphorylation level of phosphosites from high to low is represented from red to blue. On the right is the KEGG and GO enrichment analysis of phosphoproteins, conducted by Fisher’s exact test method. When *p* value is less than 0.05, it is considered to be significantly enriched, and the partial enrichment results are displayed.

**Figure 5 ijms-26-03471-f005:**
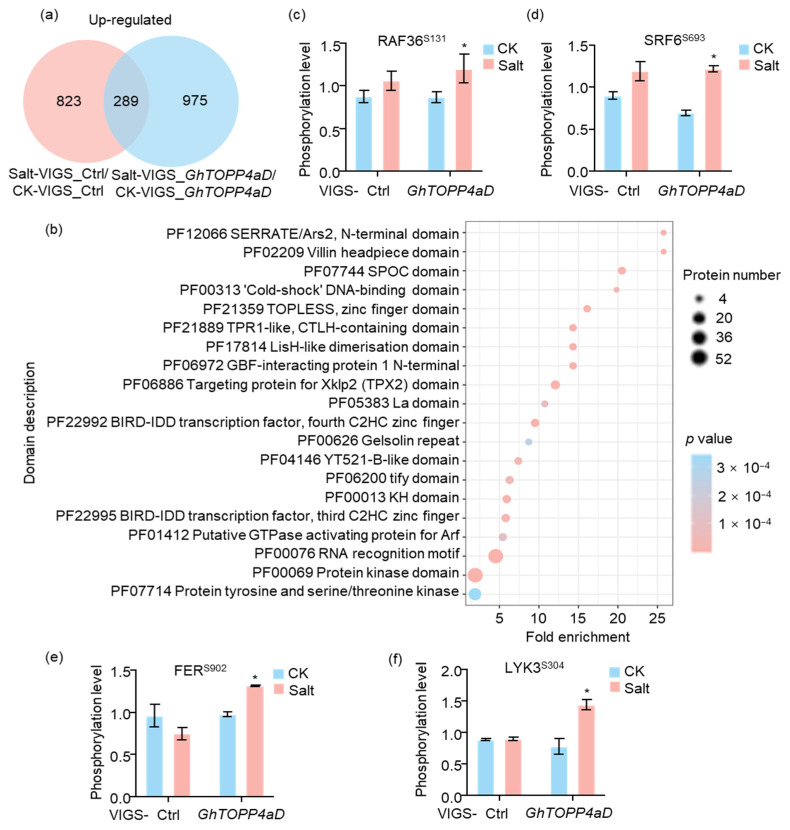
Enrichment analysis of the conserved domains in differentially upregulated phosphoproteins in response specifically to GhTOPP4aD after 300 mM NaCl. (**a**) Venn diagram shows the number of differentially upregulated phosphosites between VIGS-Ctrl and VIGS-*GhTOPP4aD* before and after salt treatment. (**b**) Domain enrichment analysis of differentially upregulated phosphoproteins in response specifically to GhTOPP4aD after 300 mM NaCl. (**c**–**f**) The phosphorylation level of phosphoprotein RAF36 (**c**), SRF6 (**d**), FER (**e**), and LYK3 (**f**) containing protein tyrosine and serine/threonine kinase domain. The data are shown as means ± SD from independent repeats (n = 2; *, *p* < 0.05, Student’s *t*-test).

**Figure 6 ijms-26-03471-f006:**
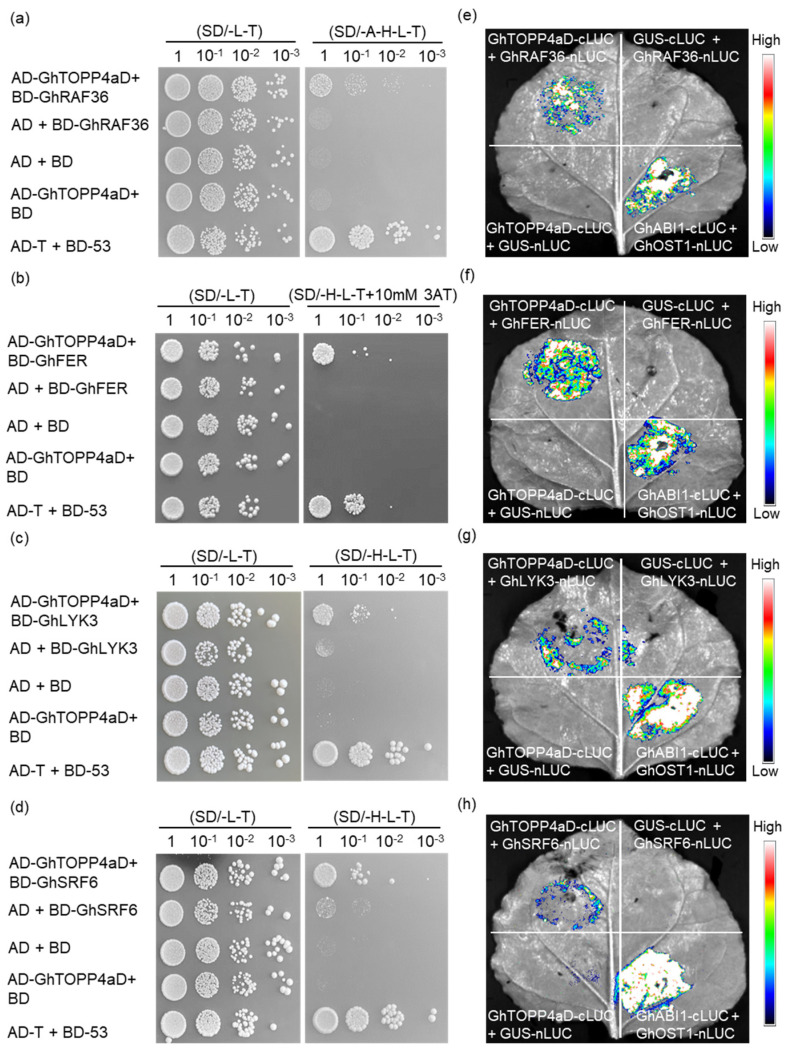
Analysis of the interaction between GhTOPP4aD and tyrosine kinases using yeast two-hybrid (Y2H) and luciferase complementation imaging (LCI) assays. (**a**–**d**) Yeast two-hybrid (Y2H) assay showing the interaction between GhTOPP4aD and tyrosine kinases. SD/-L-T, synthetic medium without Trp and Leu; SD/-H-L-T, synthetic medium without Trp, Leu, His; SD/-A-H-L-T, synthetic medium without Trp, Leu, His, Ade. DNA binding domain (BD) and activation domain (AD) were used as empty controls. (**e**–**h**) LUC complementation imaging (LCI) assay was used to assess the binding between GhTOPP4aD and tyrosine kinases. Representative images of *N. benthamiana* leaves 48 h after infiltration are shown. The bar showing red to blue indicates luciferase signal intensity from high to low.

**Figure 7 ijms-26-03471-f007:**
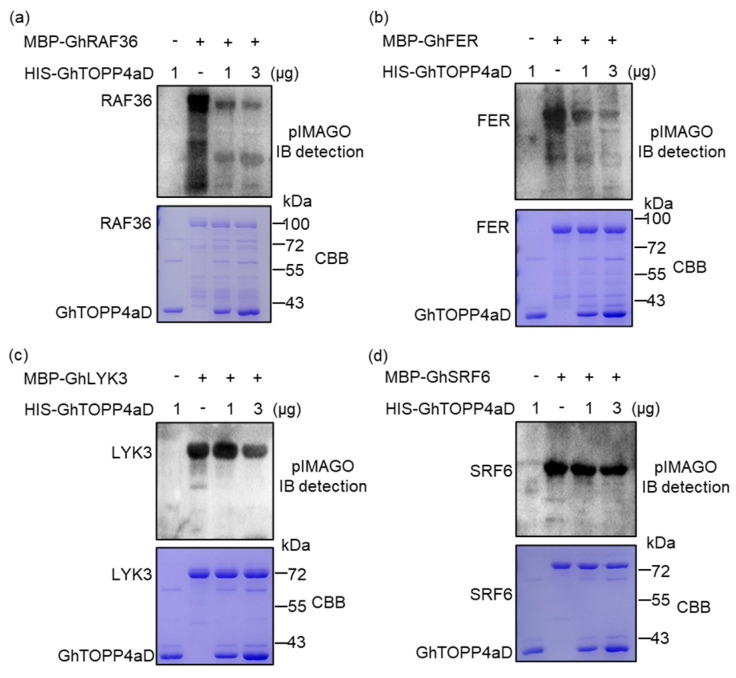
Western-blot analysis of the dephosphorylation regulatory relationship of GhTOPP4aD towards tyrosine kinases through dephosphorylation assays in vitro. (**a**–**d**) Analysis of the phosphorylation status of tyrosine kinases through in vitro dephosphorylation assays. Recombinant proteins of MBP-GhRAF36, MBP-GhFER, MBP-GhLYK3, and MBP-GhSRF6 were incubated with a gradient concentration of His-GhTOPP4aD for the dephosphorylation assay at 30 °C for 30 min and separated by 12.5% SDS-PAGE, respectively.

**Figure 8 ijms-26-03471-f008:**
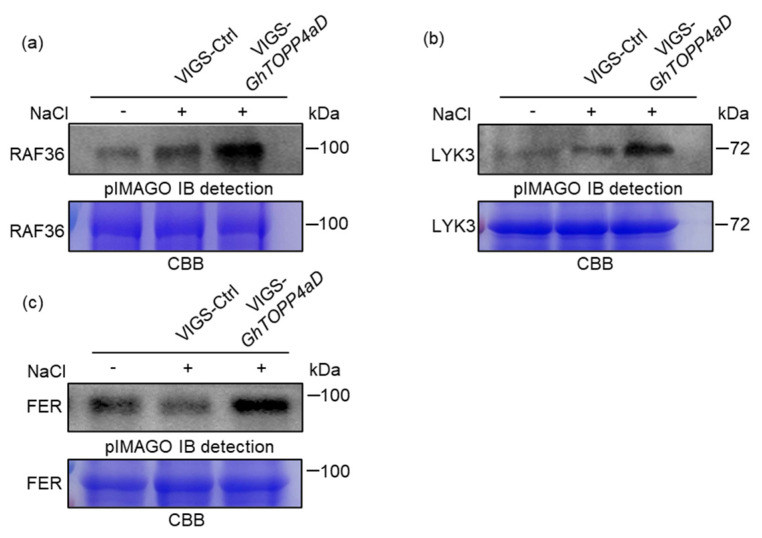
GhRAF36, GhLYK3, and GhFER kinase activity is enhanced in VIGS-*GhTOPP4aD* plants. (**a**–**c**) The phosphorylation level of GhRAF36, GhLYK3, and GhFER increases in VIGS-*GhTOPP4aD* plants. The total proteins were extracted as kinases from leaves of fourteen-day-old VIGS plants with 300 mM NaCl treatment. Pre-dephosphorylated MBP-GhRAF36, MBP-GhLYK3, and MBP-GhFER with Calf Intestine Phosphatase Alkaline (CIPA) as a substrate were incubated with total proteins in kinase reaction buffer for 30 min at 30 °C and then separated by SDS-PAGE.

**Figure 9 ijms-26-03471-f009:**
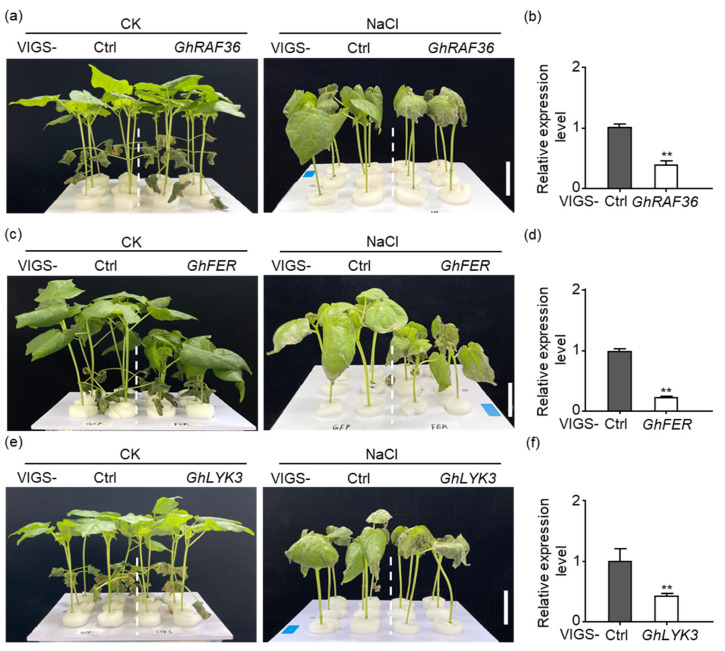
Salt stress-related phenotypes in VIGS-*GhRAF36*, VIGS-*GhFER*, and VIGS-*GhLYK3* plants. (**a**–**f**) The phenotypes of VIGS plants with or without NaCl treatment and their corresponding relative expression levels. “CK” represents control plants without NaCl stress while “NaCl” represents plants with 300 mM NaCl treatment. The dashed lines separate different VIGS groups. Bar = 3 cm. The leaf samples from VIGS plants were collected to detect the expression of *GhRAF36* (**b**), *GhFER* (**d**), and *GhLYK3* (**f**) without NaCl treatment by real-time quantitative PCR (RT-qPCR). *GhActin9* was used as the internal control. The data are shown as means ± SD from three independent repeats (n = 3; **, *p* < 0.01, Student’s *t*-test).

**Table 1 ijms-26-03471-t001:** MS/MS spectrum database search analysis summary.

Title	Number
Total spectrums	158,227
Matched spectrums	44,865
Peptides	14,991
Modified peptides	13,599
Identified proteins	6810 (6055)
Comparable proteins	6048 (5597)
Identified sites	17,919 (12,608)
Comparable sites	13,126 (10,939)

The numbers in brackets represent the number of sites or proteins with localization probability more than 0.75.

**Table 2 ijms-26-03471-t002:** The differentially phosphorylated proteins identified under NaCl treatment mentioned in this study.

Accession	Gene	ID	Description	Sequence	Phos-Site
A0A1U8IDN3	*GhABI5*	Gh_D09G1317	protein ABSCISIC ACID-INSENSITIVE 5-like	TIGGVAPVSPVSSER	S309
A0A1U8JCI3	*GhRAF36*	Gh_D05G1535	Group C Raf-like protein kinase RAF36	SGSPAPSSISSPLR	S131
A0A1U8K1V6	*GhFER*	Gh_D11G1618	Receptor-like protein kinase FERONIA	AVFSEIR	S902
A0A1U8NWI1	*GhLYK3*	Gh_A02G1137	LysM domain receptor-like kinase 3	KPSFCCGSGR	S304
A0A1U8JI82	*GhSRF6*	Gh_A06G1240	Protein STRUBBELIG-RECEPTOR FAMILY 6	TIGTDQGASPR	S693

## Data Availability

The data disclosed in this study can be accessed through the article or Appendix A provided here.

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
