# Peer review of "Type One Protein Phosphatase 4aD Negatively Regulates Cotton (Gossypium hirsutum) Salt Tolerance by Inhibiting the Phosphorylation of Kinases That Respond to Abscisic Acid"

_ijms, 2025, doi:10.3390/ijms26083471_

Round 1

Reviewer 1 Report

Comments and Suggestions for Authors

The findings in the manuscript titled: GhTOPP4aD negatively regulates cotton (Gossypium hirsutum) salt tolerance by inhibiting the phosphorylation of kinases that respond to ABA by Cao et al. is a high quality and high-impact manuscript. The manuscript is very well-written with few grammatical issues that can be easily rectified.  The authors have very conclusively shown that GhTOPP4aD phosphatase physically interacts and regulates de-phosphorylation of three major tyrosine kinases like FER, RAF36, and LYK3 kinases, which in turn, confers cotton salt- tolerance. The authors have done a very detailed, thorough research work using appropriate biochemical and molecular and biochemical strategies (more importantly the right sets of positive and negative controls in the molecular experiments have been employed in the work) to identify the tyrosine kinases and ABI5 protein that are de-phosphorylation targets of GhTOPP4aD. This research has laid the foundation for future research that will confirm if GhTOPP4aD directly de-phosphorylates Ser309 on ABI5 (induced by ABA) and the kinase/s responsible for phosphorylation of Ser309. Additionally, findings from this manuscript can be now used to probe the functional role of GhTOPP4aD and GhRAF36 in the ABA-induced signal transduction pathway which confers enhanced resistance to salt stress in cotton. In summary, this research will prove to be crucial for plant breeders for identification of molecular candidates that can be genetically engineered to generate and breed salt-tolerant cotton lines.

I have a few scientific questions and suggestions for the authors that can improve the quality of the manuscripts. I have highlighted these issues and have added my comments in the manuscript pdf file. Please see the attached file.

Other comments:

  1. In the supplementary Figure file, I have one question: Why did the authors use MEGA 6 for the generation of the phylogenetic tree when MEGA 12 is available in 2025?
  2. Also all graphs or Figure legends should clearly state what “CK” label means (although I understand this represents plants that were not stressed [control plants].
  3. GhTOPP4aD is the wild type gene name and it should not be italicized. It has been italicized throughout the text in the manuscript. GhTOPP4aD mutant plants should be italicized. Please let me know if the authors have any questions about my comments.

Comments on the Quality of English Language

Minor English problem is evident in the manuscript. Please read my comments in the manuscript pdf file to rephrase specific sentences that can improve the quality of the manuscript.

Reviewer 2 Report

Comments and Suggestions for Authors

This topic provides important insights into understanding the abiotic stress tolerance mechanism in essential cash crop (Cotton). In general, I recommend accepting the manuscript after minor revisions, including the following:

  1. The authors should include a list of abbreviations and their definitions after the abstract section. Some abbreviations, including those in the title, are not defined, which causes confusion.
  2. The captions of all figures should be more concise. I suggest transferring some details to the methodology section.
  3. I recommend modifying the keywords to be a little different from the words in the title, which can enhance and broaden the paper’s reach.
